# Novel Resistance Regions Carrying Tn*aphA6*, *bla*_VIM-2_, and *bla*_PER-1_, Embedded in an IS*Pa40*-Derived Transposon from Two Multi-Resistant *Pseudomonas aeruginosa* Clinical Isolates

**DOI:** 10.3390/antibiotics12020304

**Published:** 2023-02-02

**Authors:** Romina Papa-Ezdra, Nicolás F. Cordeiro, Matilde Outeda, Virginia Garcia-Fulgueiras, Lucía Araújo, Verónica Seija, Juan A. Ayala, Inés Bado, Rafael Vignoli

**Affiliations:** 1Departamento de Bacteriología y Virología, Instituto de Higiene, Facultad de Medicina, Av. Alfredo Navarro 3051, Montevideo 11600, Uruguay; 2Departamento de Laboratorio Clínico, Área Microbiología, Hospital de Clínicas, Facultad de Medicina, Universidad de la República, Av. Italia s/n, Montevideo 11600, Uruguay; 3Centro de Biología Molecular “Severo Ochoa” (CBMSO)-CSIC, C. Nicolás Cabrera 1, 28049 Madrid, Spain

**Keywords:** *Pseudomonas aeruginosa*, *bla*
_VIM-2_, *bla*
_PER-1_, transposon

## Abstract

Antibiotic resistance is an alarming problem throughout the world and carbapenem-resistant *Pseudomonas aeruginosa* has been cataloged as critical in the World Health Organization list of microorganisms in urgent need for the development of new antimicrobials. In this work, we describe two novel resistance regions responsible for conferring a multidrug resistance phenotype to two clinical isolates of *P. aeruginosa* (Pa873 and Pa6415) obtained from patients hospitalized in the ICU of University Hospital of Uruguay. Bacterial identification and antibiotic susceptibility tests were performed using MALDI-TOF and the Vitek 2 system, respectively. WGS was performed for both isolates using Oxford Nanopore Technologies and Illumina and processed by means of hybrid assembly. Both isolates were resistant to ceftazidime, cefepime, piperacillin–tazobactam, aztreonam, and imipenem. Strain Pa6415 also showed resistance to ciprofloxacin. Both strains displayed MICs below the susceptibility breakpoint for CAZ-AVI plus 4 mg/L of aztreonam as well as cefiderocol. Both resistance regions are flanked by the left and right inverted repeats of IS*Pa40* in two small regions spanning 39.3 and 35.6 kb, for Pa6415 and Pa873, respectively. The resistance region of Pa6415 includes Tn*aphA6*, and the new Tn*7516* consists of IRi, In899, *qacEΔ1-sul1*-IS*CR1*, *qnrVC6*-IS*CR1*-*bla*_PER-1_-*qacEΔ1-sul1*, *araJ*-like, IS*481*-like *tnpA*, IS*Pa17*, and IRR. On the other hand, the resistance region of Pa873 includes Tn*aph6* and the new Tn*7517* (IRi, In899, *qacEΔ1-sul1*, IS*CR1*–*bla*_PER-1_–*qacEΔ1-sul1*, *araJ*-like, IS*481*-like *tnpA*, IS*Pa17*, and IRR). It is necessary to monitor the emergence of genetic structures that threaten to invalidate the available therapeutic resources.

## 1. Introduction

Antibiotic resistance is an alarming problem throughout the world that permanently challenges the available therapeutic resources, both old and new. Gram-negative bacilli have taken the forefront of this problem, and carbapenem-resistant *Pseudomonas aeruginosa* has been cataloged as critical in the World Health Organization list of microorganisms in urgent need for the development of new antimicrobials [1].

Although *P. aeruginosa* is considered a ubiquitous and opportunistic pathogen, it is one of the most frequent agents of nosocomial infections in intensive care units, and in immunocompromised and burn patients [2,3].

Regardless of the resistance mechanisms involved, the term *P. aeruginosa* with “difficult-to-treat” resistance has recently been coined for those isolates that are non-susceptible to all of the following antibiotics: piperacillin–tazobactam, ceftazidime, cefepime, aztreonam, meropenem, imipenem–cilastatin, ciprofloxacin, and levofloxacin [4].

However, in the fight against antimicrobial resistance, metallo-β-lactamase (MBL)-producing *P. aeruginosa* is one of our main Achilles heels, since it is capable, from the outset, of avoiding the action of last-resource therapeutic options such as ceftolozane–tazobactam, ceftazidime–avibactam, and even cefiderocol [5,6]. The most frequently described MBLs in this species is VIM-2, followed by the IMP, NDM, and SPM variants, and the clinical problem is increased if these resistance determinants are produced by epidemic high-risk clones such as ST235, ST111, ST233, ST244, ST357, ST308, ST175, ST277, ST654, and ST298 [3]. The gene *bla*_VIM-2_ is often embedded in the variable region of class 1 integrons, both in *P. aeruginosa* and other *Pseudomonas* species, including *P. putida*, where it can be co-transferred with other genes conferring resistance to different antibiotic classes [7,8].

Briefly, class 1 integrons are typically constituted by two conserved segments named 5′-CS and 3′-CS, flanking a variable region. The 5′-CS segment includes (i) the class 1 integrase-coding gene (*intI1*), which, unlike the remaining integron components, is located in the complementary DNA strand and whose product is a site-specific recombinase that catalyzes the insertion and excision of gene cassettes; (ii) a promoter Pc that allows the inserted genes to be expressed; and (iii) the *attI* recombination site. The 3′-CS region usually comprises a truncated version of the quaternary ammonium compound resistance gene, *qacE1* (*ΔqacE1*), and the sulfonamide resistance gene, *sul1*. Finally, the variable region comprises one or more gene cassettes and their respective recombination sites, *attC* or 59-be. Gene cassettes comprise one or two genes, usually without promoter, and the *attC* site, which can exist as free non-replicative circular forms or incorporated into integrons [9,10]. Additionally, in Tn*402*-associated class 1 integrons, downstream of *ΔqacE1-sul1*, there is a Tn*402 tni* module (*tniABQC*) that makes its self-transferability possible [7,11]. Finally, complex class 1 integrons emerge by insertion downstream of 3′-CS in circles containing an IS*91*-related transposase named IS*CR1* and its associated resistance genes, generating a second variable region and a partial duplication of 3′-CS [10,12]. Various ESBL-coding genes are frequently mobilized by such structures; among them, the *bla*_CMY_ and *bla*_CTX-M_ [12] variants and *bla*_PER-1_ have been reported [13].

Although MBLs do not confer resistance to aztreonam, the co-expression of these enzymes along with other β-lactamases such as ESBLs also invalidates this antibiotic [14]. Among the most frequently detected ESBLs in *Pseudomonas* spp. are *bla*_CTX-M_, *bla*_PER_, and *bla*_GES_. In particular, *bla*_PER-1_ constitutes a significant problem on account of conferring relative resistance to β-lactam/β-lactamase inhibitor combinations such as ceftolozane–tazobactam and ceftazidime–avibactam [15]. Reports of *P. aeruginosa* co-producing both *bla*_VIM-2_ and *bla*_PER-1_ are scarce, being previously described in Italy, Turkey, and Uruguay, but the information regarding their shared genetic context was not explored, and to the best of our knowledge, there are no reports so far describing complex integrons harboring both *bla*_VIM-2_ and *bla*_PER-1_ in a single structure [16].

Among other antibiotics available for the treatment of *P. aeruginosa* infections, aminoglycosides such as gentamicin, tobramycin, and amikacin are frequently used, but the wide dissemination of acquired resistance genes coding for aminoglycoside-modifying enzymes (AMEs) among clinical isolates has been menacing its efficacy [17]. AMEs mediate aminoglycoside inactivation by catalyzing the modification of hydroxyl or amino groups by means of acetylation (AAC), phosphorylation (APH), or adenylation (ANT), where AAC variants are generally the most frequently encoded by *P. aeruginosa*, whereas APH and ANT are more common in *Acinetobacter baumannii* [18]. In *P. aeruginosa*, the AMEs more often found are AAC(6′)-Ib and ANT(2″)-I, both conferring resistance to gentamicin and tobramycin; meanwhile, resistance to amikacin can be mediated by APH(3′)-VI, which is more commonly associated with *A. baumannii* [17,18]. In the latter, *aph(3′)-VIa* has been described to be included in a transposon named Tn*aphA6*, flanked by two copies of IS*Aba125*, which has not been reported so far in *P. aeruginosa* [18,19].

In the present work, we describe two IS*Pa40*-derived resistance regions harboring Tn*aphA6, bla*_VIM-2_, and *bla*_PER-1_, with the latter two being in two different class 1 complex integrons; both genetic structures were detected in two unrelated clinical isolates of *P. aeruginosa* with high-level resistance to ceftazidime, aztreonam, and amikacin.

## 2. Results

### 2.1. Antibiotic Susceptibility Testing

*P. aeruginosa* Pa6415 and Pa873 were isolated from cerebrospinal fluid in November 2016 and tracheal secretions in October 2021, respectively. Both isolates were resistant to ceftazidime, cefepime, piperacillin–tazobactam, aztreonam, and imipenem. Strain Pa6415 also showed resistance to ciprofloxacin (MIC = 1 mg/L) but showed susceptibility to meropenem (MIC = 1 mg/L), ceftazidime–avibactam (MIC = 4 mg/L), and gentamicin (MIC = 4 mg/L). Conversely, Pa873 showed intermediate levels of resistance to meropenem (MIC = 4 mg/L) and displayed susceptibility towards ciprofloxacin (MIC = 0.125 mg/L) and gentamicin (MIC ≤ 1 mg/L). Additionally, both were susceptible to cefiderocol (Table 1).

The determination of susceptibility to CAZ-AVI in Mueller–Hinton supplemented with 4 mg/L of aztreonam resulted in a two-fold decrease in the MICs compared with the values of CAZ-AVI alone, with both falling under the susceptibility breakpoint.

### 2.2. Genetic Features of Pa6415

The in silico MLST analysis of Pa6415 agreed with previous results that it belongs to sequence type ST463 (16). The isolate also was found to belong to serotype O4 and to carry more than 200 genes related to virulence. Among the virulence determinants, we highlight the presence of both *exoS* and *exoU* genes (*exoS+*/*exoU+*), which are effectors of a type III secretion system; the exotoxin A-coding gene, *exoA*; the elastase-coding gene, *lasB*; and several alleles of genes related to alginate (*alg* and *muc*), flagella (*fle*, *flg*, *fli* and *mot*), type IV pili (*pil*), phenazine (*phz*), and rhamnolipid (*rhl*) biosynthesis among others.

Pa6415 harbors 14 resistance genes conferring resistance to six antibiotic families: aminoglycosides (*aph(3′)-VIa*,* aph(3″)-Ib*,* aph(6)-Id*, and* aph(3′)-IIb*); β-lactams (*bla*_PER-1_, *bla*_PAO_, and *bla*_OXA-50-like_) including carbapenems (*bla*_VIM-2_); fluoroquinolones (*qnrVC6*); fosfomycin (*fosA*); chloramphenicol (*catB7*); and sulfonamides (*sul1*). Additionally, we detected a β-lactamase pseudogene (*bla*_TEM-1B_) featuring a truncated initiation codon due to the insertion of IS*Aba125* (Figure 1).

Among such genes, those conferring resistance to clinically relevant anti-*P. aeruginosa* antibiotics (i.e., oxyimino-cephalosporins, carbapenems, aminoglycosides, and fluoroquinolones) are located in a 39.3 kb resistance region characterized by the insertion of multiple mobile genetic elements into one belonging to the Tn*3* family (IS*Pa40*), which will be described below.

### 2.3. Genetic features of Pa873

On the other hand, strain Pa873 belongs to ST395 and serotype O6. As found in Pa6415, this strain also carries several virulence-coding genes related to type III secretion system expression; elastase; exotoxin; and alginate, flagella, type IV pili, phenazine, and rhamnolipid biosynthesis. Among them, we note the presence of *exoA*, *lasB*, and *exoS* but not *exoU* (*exoS+*/*exoU−*).

Pa873 carries 11 resistance genes covering five antibiotic families: aminoglycosides (*aph(3′)-VIa* and *aph(3′)-IIb*); β-lactams (*bla*_PER-1_, *bla*_PAO_, and *bla*_OXA-50-like_) including carbapenems (*bla*_VIM-2_); fosfomycin (*fosA*); chloramphenicol (*catB7*); and sulfonamides (*sul1*). Like strain Pa6415, the main antibiotic resistance genes are located in a 35.6 kb IS*Pa40*-related resistance region.

### 2.4. Resistance Regions and New Transposition Units (Tn7516 and Tn7517)

Both resistance regions are depicted in Figure 1 and are bounded by IS*Pa40* inverted left (5′-GGGGAGCCCGCAGAACTCGGAAAAAATCGTACGCTAAGGTTTTCCGAGC-T) and right repeats (5′-CACATGGCGCGGCTTAGCGTACGATTTTTTCCGAATTC-TGCGGGCACCCA) (resistance regions sequences are available in Appendix A and deposited in GenBank under acc. Nos. OP329418 for Pa873 and OP329419 for Pa6415).

Although the complete IS*Pa40* transposase gene can be found in both resistance regions, it is twice truncated by the insertion of two other transposons, namely, IS*Pu23* (flanked by the direct repeats 5′-GGCC) and Tn*2* (flanked by the direct repeats 5′-GTGTT). The latter carries *bla*_TEM-1b_ as an accessory gene, albeit with a deleted start codon (ATG) on account of the insertion of the Tn*aphA6* complex transposon. This transposon consists of two directly oriented copies of IS*Aba125* flanking an *aph(3′)-VIa* gene that encodes an aminoglycoside-(3′)-phosphotransferase, accounting for the amikacin-resistant phenotype displayed by the two isolates herein reported. Since Tn*aphA6* is inserted in *bla*_TEM-1b_, the overall structure of Tn*2* (inverted repeats, transposase, and resolvase) remains unchanged, suggesting that *aph(3′)-VIa* could be mobilized either by Tn*aphA6* or along with Tn*2*. Downstream of this structure, a third fragment of IS*Pa40* can be found, consisting of 1,677 bp of the *tnpA* gene, the complete *tnpR* resolvase gene, and the *res III* and *res II* sequences.

Adjacent to IS*Pa40 tnpR*, there are segments of 19.8 kb and 16.1 kb in Pa6415 and Pa873, respectively, flanked by IRi (5′-TGTCGTTTTCAGAAGACGGCTGCAC) and IRt sequences (5′-GTGCAGTCGTCTTCTGAAAATGACA), which represent truncated Tn*402*-associated class 1 integrons [11]. In Pa6415, such structure is constituted by the following elements: a complex class 1 integron derived from In899, with *bla*_VIM-2_ as the only gene cassette, followed by IS*CR1* and the quinolone resistance gene (*qnrVC6*), and a second copy of IS*CR1* followed by the ESBL-coding gene (*bla*_PER-1_). Thus, the overall gene arrangement of the integron is *intI1*–*bla*_VIM-2_–*qacEΔ-1*-*sul1*–IS*CR1*–*qnrVC6*–IS*CR1*–*bla*_PER-1_–*qacEΔ1-sul1*. Further downstream, instead of the expected Tn*402*-related 4,630 bp *tniCQBA*-IRt module, we detected a 3,626 bp gene coding for an AraJ-like MFS-family transporter, followed by a Tn*481*-family transposase gene and a 1,035 bp remnant of Tn*402*-related *tniA*-IRt. Conversely, the resistance region in strain Pa873 features a similar structure, albeit without the IS*CR1*-*qnrVC6* tandem. Next to such platform, both strains feature IS*Pa17* delimited by its corresponding IRL (5′-TGTCATTTTCAGAAGACGGCTGCAC) and IRR (5′-GTGCAGTCGCCTTCTGAAAACGACA). The presence of IS*Pa17* adjacent to the truncated Tn*402*-associated class 1 integron and the nucleotide identity between the IRi of Tn*402* and the IRL of IS*Pa17* suggest that the whole structure, delimited by the IRi of Tn*402* and the IRR of IS*Pa17*, could function as a IS*Pa17*-based transposition unit, where the IS*Pa17* transposase would be responsible for such mobilization. Furthermore, the 5′-CGCAG-3′ direct repeats were detected immediately upstream of IRi and downstream of IRR, supporting the theory that these structures could move as a unit. Consequently, these IS*Pa17*-based putative transposition units were designated as Tn*7516* and Tn*7517* in isolates Pa6415 and Pa873, respectively [20].

In summary, the 22,020 bp Tn*7516* consists of IRi, In899, *qacEΔ1-sul1-ISCR1*, *qnrVC6*-IS*CR1*-*bla*_PER-1_-*qacEΔ1*-*sul1*, *araJ*-like, IS*481*-like *tnpA*, Tn*402* Δ*tniA*-IRt, IS*Pa17*, and IRR. On the other hand, the 18,371 bp-spanning Tn*7517* comprises IRi, In899, *qacEΔ1-sul1*, IS*CR1*–*bla*_PER-1_–*qacEΔ1-sul1*, *araJ*-like, IS*481*-like *tnpA*, Tn*402* Δ*tniA*-IRt, IS*Pa17*, and IRR. Downstream of both transposons, the resistance region ends with the remaining 2884 bp of IS*Pa40*, which include the *chrB1-chrA-sod* genes and its left inverted repeat.

### 2.5. BLAST Analysis

As mentioned above, both Pa6415 and Pa873 resistance regions are composed of various genetic elements embedded in IS*Pa40*. The BLAST analysis of these structures in the GenBank database revealed the presence of novel arrangements.

Tn*aphA6* is a transposon widely found in *Acinetobacter* spp., but when restricted to *Pseudomonas* spp., the BLAST analysis demonstrated that there are no previous reports of this transposon in such genera. However, there are previous descriptions of either *aphA6* or IS*Aba125*, the last only in ten *P. aeruginosa* isolates also harboring the metallo-β-lactamase-coding gene, *bla*_NDM-1_.

On the other hand, the presence of In899 adjacent to IS*CR1* conforming a complex class 1 integron has not been previously reported in GenBank, thus resulting in a novel platform. Moreover, the search for both *bla*_VIM-2_ and *bla*_PER-1_ resulted in a single match of a *Pseudomonadaceae* plasmid (GenBank acc. No. CP113227) harboring IS*CR1*–*qnrVC6*–IS*CR1*–*bla*_PER-1_ in an platform identical to that described in our work, but, upstream, associated with a different class 1 integron (*intI1*–*aac(6′)-Ib4*–*aadA1*–*qacEΔ1-sul1*) (Figure 2) and *bla*_VIM-2_ embedded in the *intI1*–*dfrB1*–*aac(6′)-Ib*–*bla*_VIM-2_–*tniC*–*tniQ*–*tniB*–*tniA* transposition module. Both platforms are spaced at a distance of more than 80 kb.

Finally, the arrangement containing IS*CR1*–*bla*_PER-1_–*gst*–*abct*–*qacEΔ1-sul1* has been widely reported in both plasmids and chromosomes of different Gram-negative rods, mainly *Pseudomonas* spp. but also *Acinetobacter* spp., *Vibrio* spp., *Aeromonas* spp., and *Enterobacterales*, among others. The presence of the IS*CR1*-*qnrVC6* module upstream of such platform is less frequent, with only four occurrences in GenBank, including arrangements with transposition units associated with class 1 integrons harboring *aac(6′)-Ib4*, *bla*_IMP-45_, *bla*_OXA-1_, and *catB3* as gene cassettes, from three *P. aeruginosa* plasmids (acc. Nos. CP061377, MF344570, and CP104871) and the aforementioned *Pseudomonadaceae* plasmid (acc. No. CP113227), where the platform is associated with a class 1 integron harboring *aac(6′)-Ib4* and *aadA1* as gene cassettes (Figure 2).

## 3. Discussion

A recent report estimated that, in 2019, the number of deaths attributable to antimicrobial resistance had climbed to 1.2 million persons, with *P. aeruginosa* being among the six most important agents, accountable for over 250,000 deaths. The same authors placed this pathogen in the top three Gram-negative rods related to mortality attributable to carbapenem resistance, surpassed only by *Acinetobacter baumannii* and *Klebsiella pneumoniae* [21].

*P. aeruginosa* populations behave in a non-clonal population structure, but some sequence types are well known to have successfully spread worldwide together with β-lactamase-coding genes, mainly carbapenemases. As mentioned above, the top 10 *P. aeruginosa* high-risk clones more recently reviewed are ST235, ST111, ST233, ST244, ST357, ST308, ST175, ST277, ST654, and ST298. Although ST395 (to which Pa873 belongs) is not present in such list, it has been catalogued as a high-risk clone related to multidrug-resistant/extensively drug-resistant *P. aeruginosa* [3,22]. On the other hand, ST463 (to which Pa6415 belongs) has recently been proposed as a potential high-risk clone, on account of the rapid emergence of highly virulent carbapenemase-producing *P. aeruginosa* belonging to this sequence type in China [22,23]. Pa6415 ST463 belongs to serotype O4 and is *exoS+*/*exoU+*, in accordance with previous reports [23]. Of note, the *exoU+* genotype has been associated with increased mortality in bloodstream infections [3] and with resistance to multiple antibiotics; its coexistence with *exoS* (*exoU+*/*exoS+* genotype) has been rarely reported, and it is highlighted to enhance antibiotic resistance [24].

In this work, we describe two unrelated *P. aeruginosa* isolates featuring two novel multi-resistance regions embedded in IS*Pa40*. This insertion sequence belongs to the Tn*3* family. Tn*3*-like transposons are frequently associated with antibiotic resistance genes both in *Enterobacterales* and *Pseudomonadales* [11] and feature a common structure formed by the *tnpA* and *tnpR* genes, as well as three res sites recognized by the resolvase to solve concatemers generated during the replication process [7].

Although IS*Pa40* is not one of the most frequently reported insertion sequences, recently, Brovedan et al. described the partial structure of a *bla*_VIM-2_-containing complex transposon in a *P. putida* isolate (GenBank acc. No. MZ382913.1), which partially matches IS*Pa40* [7]. This insertion sequence is interrupted by multiple transposons belonging to different families as mentioned above; nevertheless, we would like to highlight two types of structures on account of their clinical relevance and novelty: Tn*aphA6* and both transposition units Tn*7516* and Tn*7517* described in this work.

Although Tn*aphA6* is a well-known mobile genetic element associated with amikacin resistance in *Acinetobacter* spp. [19], the occurrence of *aph(3′)-VIa* in *Pseudomonas* spp. is a rare event [17]. Previous studies in our country have shown high levels of amikacin resistance in carbapenemase-producing *A. baumannii* clinical isolates (85%), whereas in *bla*_VIM-2_-bearing *P. aeruginosa* isolates, resistance to such aminoglycoside barely reached 10% [8,25]. The occurrence of Tn*aphA6* in *Pseudomonas* spp. is novel and constitutes a threat to the aforementioned therapeutic resource for these species, especially since this transposon could in theory be mobilized either independently as Tn*aphA6* or associated with the Tn*2* in which it is embedded, since both genetic structures retain their mobilization-related features.

Tn*7516* and Tn*7517* could be defined as IS*Pa17*-based transposition units that are mobilized by the transposase of such insertion sequence given the high nucleotide identity between its IRL and the IRi of the remnant of a truncated Tn*402*-associated class 1 integron. In contrast to previous reports [9,11], IS*Pa17* is downstream of the class 1 integron, but the presence of 5 bp direct repeats bracketing the whole structure supports this assumption. Similar structures have previously been related to the demobilization of other MBL-coding genes such as *bla*_IMP_, where IS*Pa17* recognizes both its IRL and IRt of Tn*402* for transposition [11].

On the other hand, the Tn*7516* and Tn*7517* transposable units carry two resistance genes that together can inactivate practically all β-lactams. In this regard, the expression of *bla*_VIM-2_ confers resistance to carbapenems, oxyimino-cephalosporins, and combinations such as ceftazidime–avibactam, whereas the expression of *bla*_PER-1_ also adds resistance to aztreonam [5,14]. Additionally, Tn*7516* carries the quinolone-resistant gene, *qnrVC6*, which could account for the ciprofloxacin resistance profile displayed by strain Pa6415 [26].

The association of *bla*_PER-1_ and *qnrVC6*, mobilized by two tandem copies of IS*CR1*, has been recently reported to be linked to a class 1 integron with a variable region constituted by *aac(6′)-Ib-IV*, *bla*_IMP-45_, *bla*_OXA-1_, *catB3* [26] or *aac(6′)-Ib4*, and *aadA1*. Nevertheless, to the best of our knowledge, this constitutes the first report of *bla*_VIM-2_ being associated with *bla*_PER-1_ and *qnrVC6* in a single transposition unit. On the other hand, the occurrence of IS*CR1*-*bla*_PER-1_-*gst*-*abct* is widely reported in GenBank and is well known to be associated with different mobile structures, such as class 1 integrons [13,27].

Keeping in mind what we mention above and the high level of structural homology between the resistance regions harboring Tn*7516* and Tn*7517*, it could be argued that IS*CR1*-*qnrVC6* can insert and excise itself independently of Tn*7517*, increasing the ability of mobilizing AMR genes on the IS*Pa40* platform [12].

Although the occurrence of *bla*_VIM-2_/*bla*_PER-1_ confers resistance to both aztreonam and ceftazidime–avibactam (CAZ-AVI), both isolates showed susceptibility to CAZ-AVI plus 4 mg/L aztreonam. Interestingly, isolate Pa6415 showed susceptibility to meropenem and CAZ-AVI despite harboring such β-lactamase genes. Further studies are required to assess whether this discordance in resistance patterns obeys different levels of gene expression, mainly of *bla*_VIM-2_. However, special attention should be given to the dissemination of *bla*_PER-1_-producing *P. aeruginosa*, since it has been associated with CAZ-AVI and ceftolozane–tazobactam resistance [15,28]. A recent multicenter European study revealed that 21.4% of the *P. aeruginosa* isolates from respiratory samples produced acquired β-lactamases, among which *bla*_PER-1_ accounted for the 48.6% and was associated with resistance to both CAZ-AVI and ceftolozane–tazobactam in all cases [28]. Conversely, in Latin America, *bla*_PER-1_ has only been reported in Uruguay and Chile, as opposed to *bla*_PER-2_ [16,29,30]. This difference may respond to the limited availability of CAZ-AVI in Latin America in contrast to Europe; thus, an increase in its use could lead to selection pressure and the subsequent dissemination of *bla*_PER-1_.

As both Pa6415 and Pa873 showed susceptibility to cefiderocol and gentamicin, these two antibiotics remain as possible therapeutic options for the treatment of infections caused by these microorganisms. Nevertheless, the scarce availability of ceftazidime–avibactam, aztreonam, and cefiderocol in Latin America dramatically reduces the therapeutic options.

Considering the potential for mobilization of each resistance gene reported in the resistance regions described in this work, special attention should be paid to the individual surveillance of co-resistance patterns in VIM-2-producing *P. aeruginosa* isolates. In this regard, resistance to amikacin could be evidencing the presence of *aph(3′)-VIa*; resistance to aztreonam, the presence of *bla*_PER-1_; and resistance to ciprofloxacin, the presence of *qnrVC6*. Strangely, none of these resistance genes have yet been reported in *P. aeruginosa* in other countries in Latin America.

## 4. Materials and Methods

### 4.1. Strains, Identification, and Antibiotic Susceptibility Testing

*P. aeruginosa* Pa873 was isolated in October 2021 from a tracheal secretion from a 78-year-old woman admitted to the intensive care unit of University Hospital of Montevideo Uruguay. On the other hand, a previously reported isolate [16], Pa6415, was obtained from the cerebrospinal fluid of a 68-year-old woman in November 2016, also admitted to the ICU of the same hospital.

Bacterial identification was performed using matrix-assisted laser desorption ionization-time-of-flight (MALDI-TOF) mass spectrometry (Bruker, Billerica, MA, USA). Antibiotic susceptibility was determined using the Vitek 2 system (bioMérieux, Marcy l’Étoile, France) and interpreted according to the Clinical and Laboratory Standards Institute (CLSI) guidelines (2022). The minimum inhibitory concentration (MIC) of ceftazidime–avibactam (CAZ-AVI) was performed with E-test (bioMérieux, Marcy l’Étoile, France) according to the manufacturer’s recommendations. The MIC of aztreonam was determined using the agar dilution method, and the MIC of CAZ-AVI plus aztreonam was determined by placing a CAZ-AVI E-test strip on a Mueller–Hinton plate supplemented with 4 mg/L of aztreonam. Values obtained for CAZ-AVI strips were compared to CAZ-AVI plus aztreonam, and both were interpreted using the CLSI breakpoints. Susceptibility to cefiderocol was determined using the disc diffusion test.

The double-disk synergy test (DDST) with combinations of antimicrobial agents and specific inhibitor disks was performed for the phenotypic detection of MBLs, class A carbapenemases, and ESBLs as previously described [31,32].

### 4.2. Short- and Long-Read Genome Sequencing

Genomic DNA was obtained using an NZY microbial genomic DNA (gDNA) isolation kit, following the manufacturer’s instructions (NZYTech Genes & Enzymes, Lisbon, Portugal). DNA quality and quantity were assessed with a NanoDrop 1000 spectrophotometer (Thermo Fisher Scientific, Wilmington, DE, USA). DNA was also quantified with Qubit^®^ 3.0 Fluorometer using Qubit^®^ dsDNA HS Assay Kit.

Libraries from gDNA were prepared using a Nextera XT kit (Illumina Inc., San Diego, CA, USA), and next-generation sequencing was performed using Illumina MiniSeq with a MiniSeq high-output reagent kit (Illumina Inc., San Diego, CA, USA) and a 2 × 151 bp paired-end approach. Reads were assembled with SPADES ver. 3.11 using k-mers 21, 33, 55, 77, 99, and 127 with the “careful” option turned on and the following cutoffs for final assemblies: minimum contig/scaffold size = 500 bp; minimum contig/scaffold average nucleotide coverage = 10-fold.

In parallel, both isolates were sequenced using an Oxford Nanopore Technologies device. Briefly, DNA libraries were prepared using a rapid sequencing kit (SQK-RAD004) following the manufacturer’s instructions. Libraries were loaded onto R9.4.1 flow cells (FLOMIN106) and sequenced for 8 h on a MinION device (Oxford Nanopore Technologies, Oxford, UK). Basecalling was performed using Guppy with a high-accuracy model, integrated into MinKNOW ver. 4.1.22 software. The quality of the generated data was assessed with NanoPlot ver. 1.33.1 [33], and Filtlong ver. 0.2.0 (https://github.com/rrwick/Filtlong accessed on 1 September 2022) was applied to remove reads shorter than 1,000 bp and reads with a mean quality score of <93. Genome hybrid assembly, using short and long reads, was performed with Unicycler ver. 0.4.8 [34].

### 4.3. Sequence Analysis

The prediction of antibiotic resistance genes was performed using both Comprehensive Antibiotic Resistance Database (select criteria, perfect and strict; sequence quality, high quality/coverage) (https://card.mcmaster.ca/ last accessed on 1 November 2022) and the ResFinder 4.1 suite (https://cge.cbs.dtu.dk/services/ResFinder/ last accessed on 1 November 2022). The MLST of both strains was predicted using the MLST 2.0 suite (https://cge.food.dtu.dk/services/MLST/ last accessed on 1 November 2022) [35]; serotype was predicted using Past 1.0 (https://cge.food.dtu.dk/services/PAst/ last accessed on 1 November 2022); and virulence-coding genes were determined in Virulence Factors Database (VFDB) (http://www.mgc.ac.cn/VFs/ last accessed on 1 November 2022).

Complete genomes were annotated using the RAST 2.0 suite (Rapid Annotation using Subsystem Technology) [36] and manually curated with Artemis software [37]. New transposon numbers were assigned by The Transposon Registry repository (https://www.lstmed.ac.uk/technical-services/the-transposon-registry last accessed on 1 November 2022) [20].

Comparisons with publicly available sequences were performed using BLAST (http://blast.ncbi.nlm.nih.gov/ last accessed on 25 January 2023), and physical maps were generated with EasyFig 2.1 using BLAST 2.2.18 (http://mjsull.github.io/Easyfig/ last accessed on 1 September 2022).

## Figures and Tables

**Figure 1 antibiotics-12-00304-f001:**
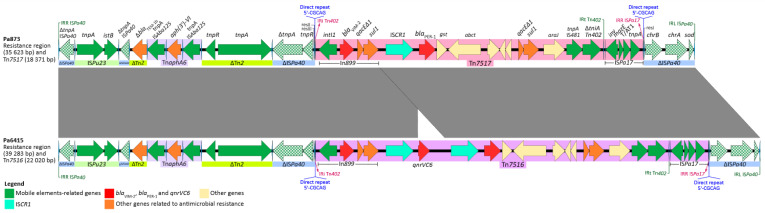
Linear map of resistance regions and Tn*7517* and Tn*7516* of *Pseudomonas aeruginosa* Pa873 (GenBank acc. No. OP329418) and Pa6415 (GenBank acc. No. OP329419), respectively. Homologous segments (≥99% identity) are shown as gray blocks. Genes are represented by arrows and colored according to their function, as shown in the legend.

**Figure 2 antibiotics-12-00304-f002:**
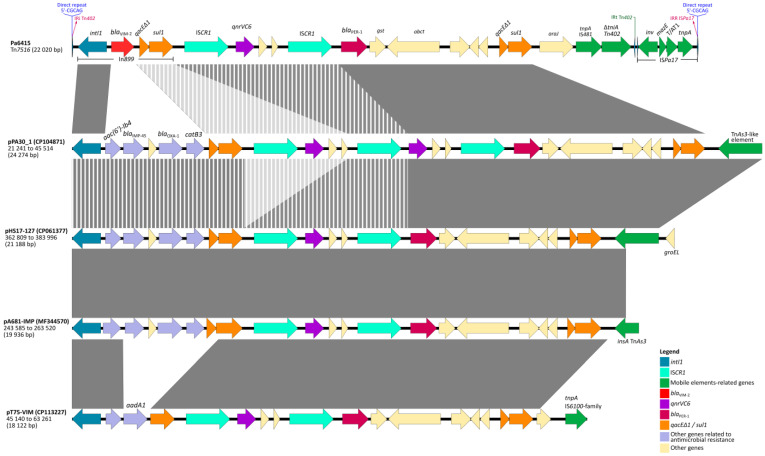
Sequence comparison (BLAST) of Tn*7516* (Pa6415) and other genetic platforms carrying *bla*_PER-1_ and *qnrVC6* obtained from GenBank. Homologous segments (>99% identity) are represented in gray bars and stripped bars when overlayed. Genes are represented by arrows and colored according to their function, as shown in the legend.

**Table 1 antibiotics-12-00304-t001:** Main features of *P. aeruginosa* isolates Pa6415 and Pa873.

	Pa6415	Pa873
Date of recovery	November 2016	October 2021
Sample origin	Cerebrospinal fluid	Tracheal aspirate
MLST	ST463	ST395
Antibiotics ^1^	Susceptibility results * (Antimicrobial resistance genes detected)	
PTZ	32 (bla_PER-1_/bla_VIM-2_)	64 (bla_PER-1_/bla_VIM-2_)
CAZ	≥256 (bla_PER-1_/bla_VIM-2_)	≥256 (bla_PER-1_/bla_VIM-2_)
FEP	≥64 (bla_PER-1_/bla_VIM-2_)	≥64 (bla_PER-1_/bla_VIM-2_)
IPM	≥32 (bla_VIM-2_)	8 (bla_VIM-2_)
MEM	1	4 (bla_VIM-2_)
GM	4	≤1
AK	≥64 (aph(3′)-VIa)	≥64 (aph(3′)-VIa)
CIP	1(qnrVC6)	0.125
ATM	>256 (bla_PER-1_)	>256 (bla_PER-1_)
CZA	4	24 (bla_VIM-2_)
CZA + ATM	1	6
FDC (mm)	26	24

^1^ Abbreviations: PTZ, piperacillin–tazobactam; CAZ, ceftazidime; FEP, cefepime; IPM, Imipenem; MEM, meropenem; GM, gentamicin; AK, amikacin; CIP, ciprofloxacin; ATM, aztreonam; CZA, ceftazidime–avibactam; CZA + ATM, CAZ-AVI plus 4 mg/L of aztreonam; FDC, cefiderocol. * MIC values are shown in mg/L.

## Data Availability

The nucleotide sequences of the resistance regions corresponding to Pa873 and Pa6415 isolates were deposited in GenBank under accession numbers OP329418 and OP329419, respectively. Until the sequences are released by the GenBank curators, we include the annotation files as Appendix A.

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
