# Peer review of "Novel Resistance Regions Carrying TnaphA6, blaVIM-2, and blaPER-1, Embedded in an ISPa40-Derived Transposon from Two Multi-Resistant Pseudomonas aeruginosa Clinical Isolates"

_antibiotics, 2023, doi:10.3390/antibiotics12020304_

Round 1

Reviewer 1 Report

Dear authors,

The article addresses a topic of wide interest: that of the molecular study of MDR strains of P.aeruginosa, which is part of the top 3 Gram negative rods related to mortality attributable to carbapenem resistance.

Advanced molecular techniques were used, but due to the small number of strains studied, further studies are required.

Please specify on the basis of which criteria the 2 strains of P.aeruginosa were chosen and which standard of interpretation of the antimicrobial sensitivity tests was used.

Author Response

Dear reviewer,

We would like to thank you for your dedication to read and review our manuscript. Your comments and suggestions where very helpful to improve our work.

Further analysis were conducted to enhance the data provided by WGS of each strain (Lines 111-116; 135-139). Also, BLAST analysis results were added, comparing the resistance regions with the GenBank database (Lines 192-222).

Regarding the selection criteria for both P. aeruginosa isolates, the resistance to imipenem, cefepime, ceftazidime, aztreonam and amikacin usually are considered as warnings about transferable mechanisms of resistance in our hospital, leading to further studies which was our case. This was clarified in the manuscript (Lines 88; 91-92).

Finally, all susceptibility results were interpreted according to the Clinical and Laboratory Standards Institute (CLSI) guidelines (Lines 321-322).

Reviewer 2 Report

Dear authors:

I read with great attention your manuscript that I find well written and the subject very interesting, and I suggest you some corrections. Plaese take the time to do them.

1. Line 40: A concise and clear research background can be presented in the section of "Introduction" by combining the content related to the same topic.

2. Line 116: It is recommended to reduce the picture to an appropriate size and place the note on the picture without covering the text content.

3. It is suggested to increase the gene homology analysis of the resistance region, comparing the similar sequences of different strains, and comprehensively explaining the resistance region sequences.

Author Response

Dear reviewer,

We would like to thank you for your dedication to read and review our manuscript. Your comments and suggestions where very helpful to improve our work.

The section “Introduction” was revised and information regarding our findings was added. We consider the content is appropriate to the research objectives and findings, suitable framing the article paper.

The picture was reduced to fit the page width, and the image description was corrected.

Gene homology analysis was added, comparing the resistance regions with the GenBank database (Lines 192-222). A new figure regarding these results was added (Figure 2).

We hope the modifications will be considered appropriate.

Best regards,

Rafael